# A Novel Spo11 Homologue Functions as a Positive Regulator in Cyst Differentiation in *Giardia lamblia*

**DOI:** 10.3390/ijms222111902

**Published:** 2021-11-02

**Authors:** Yu-Chien Chen, Szu-Yu Tung, Chia-Wei Huang, Soo-Wah Gan, Bo-Chi Lin, Chia-Wei Chen, Zi-Qi Lin, Chin-Hung Sun

**Affiliations:** Department of Tropical Medicine and Parasitology, College of Medicine, National Taiwan University, Taipei 100, Taiwan; r02445201@ntu.edu.tw (Y.-C.C.); sayeodong@gmail.com (S.-Y.T.); ji3g4cj6ru8jo3@hotmail.com (C.-W.H.); autumngsw@gmail.com (S.-W.G.); r98445203@ntu.edu.tw (B.-C.L.); dog111443@hotmail.com (C.-W.C.); 100312015@gms.tcu.edu.tw (Z.-Q.L.)

**Keywords:** cyst, Spo11, transcription, *Giardia*, differentiation, DNA binding protein, parasite, transcription regulation

## Abstract

*Giardia lamblia* persists in a dormant state with a protective cyst wall for transmission. It is incompletely known how three cyst wall proteins (CWPs) are coordinately synthesized during encystation. Meiotic recombination is required for sexual reproduction in animals, fungi, and plants. It is initiated by formation of double-stranded breaks by a topoisomerase-like Spo11. It has been shown that exchange of genetic material in the fused nuclei occurs during *Giardia* encystation, suggesting parasexual recombination processes of this protozoan. *Giardia* possesses an evolutionarily conserved Spo11 with typical domains for cleavage reaction and an upregulated expression pattern during encystation. In this study, we asked whether Spo11 can activate encystation process, like other topoisomerases we previously characterized. We found that Spo11 was capable of binding to both single-stranded and double-stranded DNA in vitro and that it could also bind to the *cwp* promoters in vivo as accessed in chromatin immunoprecipitation assays. Spo11 interacted with WRKY and MYB2 (named from myeloblastosis), transcription factors that can activate *cwp* gene expression during encystation. Interestingly, overexpression of Spo11 resulted in increased expression of *cwp1*-*3* and *myb2* genes and cyst formation. Mutation of the Tyr residue for the active site or two conserved residues corresponding to key DNA-binding residues for *Arabidopsis* Spo11 reduced the levels of *cwp1-3* and *myb2* gene expression and cyst formation. Targeted disruption of *spo11* gene with CRISPR/Cas9 system led to a significant decrease in *cwp1-3* and *myb2* gene expression and cyst number. Our results suggest that Spo11 acts as a positive regulator for *Giardia* differentiation into cyst.

## 1. Introduction

*Giardia lamblia* causes a diarrheal disease contracted by human beings from ingesting cysts in contaminated water or food [1,2]. Prevalence of giardiasis is 10-fold higher in developing countries than in developed countries [1,2]. Poor hygiene and water resource contamination can lead to fecal–oral transmission and waterborne transmission [1,3]. Symptoms can vary from asymptomatic infection and acute self-limiting diarrhea to chronic diarrhea, which leads to malnutrition and growth failure in children [4,5,6]. Some patients develop irritable bowel syndrome with long-term gastrointestinal symptoms and food allergy [2,7].

*Giardia* is a unique model as a simpler organism to investigate mitochondria relic organelles and a minimal set of components for biological pathway, as many orthologues are actually missing from *Giardia* [8]. Therefore, it is of interest to understand the differentiation mechanism of this simpler organism, which may provide a phylogenetic key to understanding the origin and evolution of various metabolism pathways. During differentiation of trophozoites into cysts, known as encystation, induction of synthesis of three cyst wall proteins (CWPs) that are transported to different vesicles than polysaccharide is a hallmark of cyst wall formation [1,9,10,11,12,13,14]. Previously, our lab identified and investigated the role of molecules involved in inducing *cwp* gene expression. These include eight transcription factors, MYB2 (named from myeloblastosis), GARP1 (named from the maize GOLDEN2, *Arabidopsis* response regulator, and *Chlamydomonas* Psr1), ARID1 (named from AT-rich interaction domain), WRKY, E2F1 (named from adenovirus E2 gene promoter binding factor), Pax1 (named from paired box), Pax2, and multiprotein bridging factor 1 (MBF1), and two topoisomerases (TOP2 and TOP3β) [15,16,17,18,19,20,21,22,23,24,25].

Spo11 is an evolutionarily conserved enzyme to create double-stranded breaks (DSBs) that help proceed meiotic recombination to generate new diversified genetic alleles in eukaryotes [26]. The Spo11 homologs have been found in animals, fungi, and plants, and also in protozoa but await to be functional characterized [27,28]. Spo11 proteins are orthologous to Archaebacteria TOP6A (Topoisomerase VIA) and have similar catalytic activity to induce DSBs [29,30,31]. TOP6 from Archaebacteria belongs to type IIB topoisomerases and contains two subunits, TOP6A and TOP6B [29]. The former contains a Toprim domain with a conserved catalytic tyrosine residue and residues for binding a metal ion cofactor and a domain with sequence similarity to *Escherichia coli* catabolite gene activator protein (CAP) DNA binding domain [26]. The latter contains an ATPase domain which is involved in ATP binding and hydrolysis [29]. Type IIB topoisomerases create a transient DSB that helps ATP-dependent crossing of two DNA duplexes, resolving topological problems during DNA replication, recombination, and transcription [32].

In mammals and fungi, only one Spo11 is present, and the known results from genetic characterization suggest the role in meiotic recombination [33]. However, three Spo11 homologues have been found in *Arabidopsis* [34]. Among the three AtSpo11 proteins, AtSpo11-1 and AtSpo11-2 proteins functions as Spo11 for meiotic recombination [27]. AtSpo11-3 can reconstitute with AtTOP6B to become a functional TOP6 enzyme [35], suggesting its role is TOP6A but not Spo11. Mutation of AtSpo11-3 and AtTOP6B resulted in dwarf phenotypes and down-regulation of many genes, suggesting that these two factors may constitute a full TOP6 enzyme that induces expression of many genes and plays a major role in somatic development [33].

Recently, in animals and fungi, orthologues for TOP6B have been found, so Spo11 has its partner to reconstituted TOP6 full enzyme for meiotic recombination [36]. Like TOP6A, the cleavage enzymatic activity of yeast Spo11 depends on the presence of Mg^2+^ [37,38]. Mutation of specific residues in the conserved Toprim domain of yeast Spo11 abolished meiotic recombination [37]. Mutation of Spo11 proteins from various organisms, including yeast, flies, nematodes, and mice, resulted in a significant defect in meiotic recombination, suggesting that these characterized Spo11 proteins are involved in meiotic recombination [31,39,40,41,42,43]. The mutation of mouse Spo11 caused infertility due to arrested spermatocyte with no synapsis, suggesting a role of Spo11 in maintaining synapsis during spermatogenesis [42,43].

Spo11 orthologues have been found in many organisms with sexual reproduction, but they have also been found in some organisms without discovered sexual reproduction or meiotic recombination [30]. Therefore, it is possible that Spo11 may have a function other than meiotic recombination. Little is known about the presence of functional Spo11 in protozoan parasites. Spo11 is expressed in the nucleus of a specific developmental form in the meiotic stage in *Trypanosoma* [44]. Upregulation of Spo11 upon starvation stage suggests its role in response to stress in *Entamoeba* [45]. It has been shown that meiosis-specific genes, including Spo11, are upregulated and expressed in nuclei and that nuclear fusion and exchange of genetic material occurs during encystation in *Giardia* [46,47]. In this study, we asked whether *Giardia* Spo11 functions in inducing *Giardia* differentiation into dormant cysts, like other topoisomerases we previously characterized [24,25]. We found that Spo11 possessed DNA-binding activity to both single-stranded and double-stranded DNA. In addition, Spo11 was capable to interact with two important encystation-induced transcription factors, WRKY and MYB2. To understand the role of Spo11 in regulating transcription, we asked whether gene expression is affected by over-expression, mutation, or targeted disruption of Spo11. We found evidence of Spo11 in inducing expression of *cwp1-3* and *myb2* genes and cyst generation, providing insights into the role of Spo11 in inducing *Giardia* differentiation into cyst.

## 2. Results

### 2.1. Analysis of the spo11 Gene

A putative topoisomerase homologue (GenBank accession number AF485825.1, open reading frame 15279) annotated as Spo11 Type II DNA topoisomerases VI subunit A was previously identified from *G. lamblia* genome database [8,25]. The deduced *Giardia* Spo11 protein is composed of 368 amino acids with a pI of 8.29 and a molecular weight of 40.72 kDa. Like *S. cerevisiae* Spo11, the *Giardia* Spo11 has a TP6A_N domain (residues 59 to 119) and a Toprim domain (residues 167 to 330) as predicted by Pfam and NCBI Conserved Domain Search (Figure 1A and Appendix A) (http://pfam.xfam.org/ accessed on 3 October 2021) (https://www.ncbi.nlm.nih.gov/Structure/cdd/wrpsb.cgi accessed on 3 October 2021) [48,49]. TP6A_N is similar to a DNA-binding domain of CAP and may be involved in DNA binding [50]. Toprim domain is topoisomerase-primase domain with a conserved active region involved in magnesium ion coordination for cleavage reaction [37].

*Giardia* Spo11 also has a conserved Tyr (residue 88), corresponding to the catalytically important Tyrosine of *S. cerevisiae* Spo11 (residue 135) (Figure 1A and Appendix A) [26]. The sequence of the CAP and Toprim domains of the *Giardia* Spo11 has some similarity to those of the Spo11 family from other eukaryotes (Appendix A). The full-length of *Giardia* Spo11 has 20.0% identity and 34.1% similarity to that of human Spo11 (calculated from Appendix A). A phylogenic tree obtained from the alignment of the Spo11 proteins from various organisms revealed that *Giardia* Spo11 (15279) is similar to Spo11 from other organisms (Appendix A). Two key contact residues identified by DNA-binding studies of *A. thaliana* Spo11 (Gly215 and Arg222) are conserved in *Giardia* Spo11 (Gly202 andArg209) (Appendix A) [51]. *Giardia* Spo11 also has the conserved Glu and two Asp (DxD) for Mg^2+^ coordination in the Toprim domain (Appendix A) [37].

### 2.2. Encystation-Induced Expression of the Spo11 Protein

It has been shown that GFP-tagged Spo11 protein is expressed only during encystation [46,47]. RT–PCR and quantitative real-time PCR analysis confirmed the increase of *spo11* mRNA (by ~9.06-fold) in 24 h encysting cells (Figure 2B). To establish the functional study of Spo11, we generated construct pPSpo11, in which the *spo11* gene is over-expressed under the control of its own promoter (Figure 1C). The HA tag (~1 kDa) was also designed to fuse to the C terminus of the protein (Figure 1C). After stable transfection with the plasmid, we found the expression of the Spo11-HA protein significantly increased during encystation (Figure 1D). 

### 2.3. Localization of the Spo11 Mutants

To further determine the function of *Giardia* Spo11, we observed the phenotype of mutation of Spo11. The Spo11 proteins possessed a catalytically important Tyr residue of the cleavage domain to create a transient DSB [26]. Therefore, we tried to mutate Tyr 88 of *Giardia* Spo11, which corresponds to Tyr 135 of *S. cerevisiae* Spo11 [26], to see if it is also important for activity (Figure 2A and Appendix A). Interestingly, the wild-type Spo11-HA was located to the nuclei and slightly to the cytoplasm (Figure 2B), although GFP-tagged Spo11 protein was previously found to localize only to nuclei during encystation [46,47]. We found that mutation of the Tyr 88 to His resulted in a decrease in nuclear localization, but an increase in cytosolic localization (Spo11m1, Figure 2C). We also found that mutation of two conserved residues of *Giardia* Spo11 (Gly202 and Arg209) corresponding to key residues for DNA-binding of *A. thaliana* Spo11 (Gly215 and Arg222) (pPSpo11m2, Figure 2A,D) decreased nuclear localization, but increased localization to cytosolic vesicles. The results suggest that Tyr88, Gly202, and Arg209 may play a role in the nuclear localization. Interestingly, in the Spo11-HA positive stained cells, the CWP1 protein was stained in the encystation secretory vesicles (ESVs), which are responsible for transportation of CWPs to cyst wall (Figure 2E) [1], suggesting that Spo11 may function in inducing the ESV and thereby in inducing cyst formation. 

### 2.4. Spo11 Positively Regulates the Expression of the cwp1-3, and myb2 Genes

Findings that the CWP1 expression in the ESVs of Spo11-HA positive stained cells suggest that Spo11 may affect encystation-induced genes. We further found that the expression of the CWP1 protein increased to a significantly higher degree in the Spo11-HA-expressing cell line than in the vector control cell line (Figure 1C and Figure 3A,C) [52]. The Spo11-HA-expressing cell line exhibited an increase in total mRNA levels of the *spo11* gene, including the endogenous *spo11* plus vector-expressed *spo11*, than the control cell line (Appendix A and Figure 3D). Interestingly, the mRNA levels of the *cwp1*-*3* and *myb2* genes were also significantly increased in the Spo11-HA-expressing cell line (Appendix A and Figure 3D). In addition, the Spo11-HA-expressing cell line exhibited a significant increase in the cyst number (Figure 3E). Similar findings were observed in encystation stage (Appendix A). Our results suggest that Spo11 can induce the *cwp1-3* and *myb2* gene expression and cyst production.

RNA-seq analysis suggests that 129 and 168 genes were significantly up-regulated (≥2-fold) and down-regulated (≤1/2) (*p* < 0.05) in the Spo11-HA-expressing cell line relative to the vector control cell line, respectively (Appendix A), although the *cwp* and *myb2* genes do not quite achieve statistical significance. 

### 2.5. Reduction of cwp Gene Expression by Mutation of Spo11

Spo11 mutants were also observed to determine the role of Spo11. The levels of Spo11m1-HA and Spo11m2-HA were similar to that of the wild-type Spo11-HA (Figure 3A,B). The Spo11m1-HA- and Spo11m2-HA-expressing cell lines displayed a significant reduction in the CWP1 level than the wild-type Spo11-HA-expressing cell line (Figure 3A–C). The mRNA levels of *spo11m1-HA* and *spo11m2-HA* were similar to that of the wild-type *spo11-HA* during vegetative growth (Figure 3D and Appendix A). We did not detect any *spo1**1-HA* transcripts in the control cell line (Appendix A). Interestingly, the Spo11m1-HA- and Spo11m2-HA-expressing cell lines exhibited a significant reduction in the mRNA levels of *cwp1-3* and *myb2* than the wild-type Spo11-HA-expressing cell line (Figure 3D and Appendix A). In addition, the Spo11m1-HA- and Spo11m2-HA-expressing cell lines displayed significant reduction in cyst number compared to the wild type Spo11-HA-expressing cell line (Figure 3E and Appendix A). We also obtained similar results with encystation samples (Appendix A). Our findings suggest a reduction of encystation-inducing activity of Spo11m1 and Spo11m2.

### 2.6. DNA Binding Activity of Spo11

Spo11 proteins have ability to bind to and cleave double-stranded DNA [26]. *Arabidopsis* SPO11-1 has ability to bind to both double-stranded and single-stranded DNA [51]. To understand whether Spo11 has DNA-binding activity, the Spo11 protein was expressed in *E. coli* and purified (Figure 4A). We performed electrophoretic mobility shift assays with purified recombinant Spo11 and pBluescript SK (+) plasmid. A smear of bound form detected in a dose-dependent manner suggests that Spo11 can bind to plasmid DNA (Figure 4B). Interestingly, Spo11 can bind to both the supercoiled and relaxed circular forms of plasmid DNA (Figure 4B). Furthermore, Spo11 can bind to linear species of plasmid DNA and PCR amplicon from the 5′-flanking region of an encystation-induced gene, *cwp1* (Figure 4C and Appendix A) [12]. Using a single-stranded DNA probe, φx174, and Spo11, we showed the formation of shifted complexes (Appendix A). Our findings suggest that Spo11 has the capacity to bind to both the double-stranded and single-stranded DNA.

### 2.7. Analysis of DNA Binding Activity of Spo11 Mutants

To understand which regions are important for Spo11 binding to DNA, we performed DNA binding assays with purified recombinant Spo11 mutants (Figure 2A). As shown in Figure 4D,E, there was no change of the DNA binding activity of Spo11m1 but an obvious decrease of DNA binding activity of Spo11m2, suggesting that Gly202 and Arg209 are important for DNA binding (Figure 2A).

### 2.8. Association of Spo11 with the cwp1-3 and myb2 Promoters In Vivo

We further used chromatin immune-precipitation (ChIP) analysis to monitor the binding of Spo11 to specific promoters in the Spo11-HA-expressing cell line in vivo. We found an association of Spo11 with the *cwp1*-*3* and *myb2* gene promoters during encystation (Figure 5A,B). Spo11 did not bind to the promoter of the U6 snRNA gene which is transcribed by RNA polymerase III, or the promoter of the 18S ribosomal RNA gene which is transcribed by RNA polymerase I (Figure 5C).

### 2.9. Spo11 Is in a Complex with WRKY or MYB2

Spo11 induces meiotic recombination by introduction of DSBs in chromosomal DNA [29,30]. Localization of DSB hotspots is usually within nucleosome-depleted regions and has connection with *cis*-acting elements and interaction of transcription factors [53]. We hypothesized that *Giardia* Spo11 may interact with transcription factors to regulate expression of *cwp* genes. We tried to test with two encystation-induced transcription factors involved in up-regulation of the *cwp* genes during encystation, WRKY and MYB2 [18,19]. We used the Spo11-HA-expressing cell line to analyze the interaction between these two transcription factors (Figure 6A,B) [18,19]. We confirmed the over-expression of the Spo11 protein and also found that Spo11 over-expression resulted in an increase in the WRKY and MYB2 protein levels (Figure 6A), but led to a decrease in the ISCS level (Figure 6A). The HA-tagged Spo11 co-immuno-precipitated with WRKY or MYB2 from encysting cell lysates, indicating that these proteins are present in a common complex in vivo (Figure 6B). As a negative control, no anti-ISCS antibody reactivity was obtained with immuno-blots of anti-HA immuno-precipitates of the pPSpo11 cell line (Figure 6B). The reciprocal immuno-precipitation also confirmed the interaction of Spo11 with WRKY or MYB2 (Figure 6C,D), suggesting that they are present in a complex. 

### 2.10. Decreased Expression of cwp1-3 and myb2 Genes by Targeted Disruption of spo11

We further investigated the role of Spo11 in encystation using our CRISPR/Cas9 system that successfully disrupted the *mlf*, *top3β*, and *mbf1* genes [23,25,54]. The CRISPR/Cas9 system was delivered by cotransfection of plasmid DNA with gRNA and Cas9 into *Giardia* and by the establishment of the stably transfected cell line with puromycin selection (Figure 7A). We verified the replacement of the *spo11* gene by the puromycin acetyltransferase (*pac*) gene using sequencing of amplified genomic DNA (Figure 7B,C and Appendix A). The *spo11* gene was partially replaced by the *pac* gene with a disruption rate of about 34% (Figure 7B,C). It has been shown that G418, a polypeptide synthesis inhibitor with cytotoxicity, can be used to test the drug sensitivity of *top3β* and *mbf1* targeted disruption cell lines [24,25]. Similarly, after G418 treatment, the Spo11td cell line exhibited a significant decrease in viability as compared to the control cell line, suggesting that Spo11td cell line displayed decreased tolerance to G418-mediated cytotoxicity (Figure 7D). In addition, we found that the Spo11td cell line exhibited a significant decrease in cyst number compared with the control cell line during vegetative growth (Figure 7E). 

We further found that the Spo11td cell line displayed a significant decrease in CWP1 protein level in comparison with the control cell line (Figure 7F). In addition, there was a significant reduction in the mRNA levels of *spo11*, *cwp1-3*, or *myb2* in the Spo11td cell line in comparison with the control cell line (Figure 7G). Similar results were obtained from the Spo11td cell line during encystation (Appendix A). We further performed subsequent analysis by the removal of puromycin. Similar results were obtained from the Spo11td –pu cell line during both vegetative growth and encystation (Appendix A). Our findings from targeted disruption of the *spo11* gene indicate that Spo11 can induce expression of *cwp1-3* and *myb2*, and cyst generation, thereby inducing encystation.

## 3. Discussion

As an orthologue of TOP6A protein family, Spo11 is capable to create DSBs that are required for meiotic recombination and chromosome segregation in most sexual eukaryotes [29,30,31]. It is hardly known whether Spo11 can modulate gene expression, although it is becoming increasingly clear that DSB repair and transcriptional activation may be highly correlated [53,55,56,57]. Increased DNA repair proteins in the transcription initiation sites suggest that they may interact with transcription factors to help transcription initiation for important genes after DSBs [55]. DSBs at gene promoters may cause transcriptional activation, possibly due to recruitment of RNA polymerase and facilitation of epigenetic modification by DNA repair proteins [56,57]. DSB localization is usually within nucleosome-depleted regions which are often found in the promoter regions during meiosis [53], suggesting that chromatin accessibility is related with DSB hotspots. 

The *Arabidopsis* AtSpo11-3 functions as TOP6A by cooperating with AtTOP6B to become a functional TOP6 enzyme [33]. Unlike Spo11, it does not function in meiotic recombination [33]. Mutation of AtSpo11-3 and AtTOP6B resulted in the identical phenotypes, cell-elongation defects throughout plant development, but no reduction in meiotic recombination [33]. In addition, these two mutants exhibited downregulation of the same set of genes, including putative cell wall-modifying proteins important for cell elongation [33], suggesting that AtSpo11-3 and AtTOP6B work in the same complex that forms TOP6. The role of AtSpo11-3 and AtTOP6B in transcriptional regulation may be due to their possible constitution of complex with chromatin-remodeling factors involved in transcriptional regulation [33]. Topoisomerases also have a role in transcription activation as recruitment of both type I and type II topoisomerases to genomic loci with higher transcriptional activity [58,59]. TopoIIβ-mediated DSBs in promoters of early response genes can be induced to switch on gene expression in neurons [60].

We found that *Giardia* Spo11 may have a similar role with topoisomerases or AtSpo11-3 in induction of transcription. *Giardia* Spo11 protein is expressed at a higher level during encystation (Figure 1B,D) [46,47], and it has DNA binding activity for single-stranded DNA and double-stranded DNA or for the *cwp1* promoter (Figure 4 and Appendix A). We hypothesize that Spo11 can form complexes with transcription factors binding to promoter regions (Figure 8). *Giardia* Spo11 may directly bind promoters, and it may also interact with other transcription factors, to regulate the *cwp* gene expression (Figure 8). RNA polymerase II was then recruited by the complex to activate *cwp* transcription (Figure 8). *Giardia* gene promoters are as short as 50 bp and contain AT-rich Inr elements for promoter activity and transcription start site selection [12,13,14,61,62,63]. In vivo association of Spo11 with the *cwp1-3* and *myb2* promoters was also confirmed. In addition, Spo11 can be co-immuno-precipitated with WRKY and MYB2. Over-expressed Spo11 increased the expression of *cwp1-3* genes by ~2.1–9.1-fold in vegetative trophozoites (Figure 3D), which is lower than ~47-fold induction of the *cwp1* promoter during encystation [16], suggesting that interaction between Spo11 and encystation-induced transcription factors is important for promoter activity. Over-expressed Spo11 did not increase the *ran* gene expression (Figure 3D), possibly due to the absence of cooperation with the encystation-induced transcription factors.

WRKY and MYB2 both bind to positive *cis*-acting elements and positively regulate *cwp* during encystation [15,19]. The presence of the WRKY and MYB2 binding sequences in the *spo11* promoter (Appendix A) [15,19] suggests that *spo11* gene expression is up-regulated by WRKY and MYB2 and that Spo11 might play a positive role in *Giardi*a encystation. Since over-expressed Spo11 increased the WRKY and MYB2 level (Figure 6A), there is a positive regulation cycle between Spo11 and WRKY or MYB2. Similarly, the *top2* and *top3β* promoters upregulated during encystation also contain the MYB2 binding sequence [24,25]. Over-expressed TOP2 and TOP3β can also upregulate the *myb2* gene [24,25]. 

Studies suggest that the two key contact residues of *Arabidopsis* Spo11 (Gly215 and Arg222) are important for DNA binding (Appendix A) [51]. We found that *Giardia* Spo11 also has these two residues (Gly202 andArg209), and it has the predicted TP6A_N domain and Toprim domain similar to that of other Spo11 family members (Figure 1A and Appendix A) [37,50]. *Giardia* Spo11 also has a conserved Tyr (residue 88), corresponding to the tyrosine of *Saccharomyces* Spo11 (residue 135) in the active center, which is involved in DNA cleavage and meiotic recombination (Figure 1A and Appendix A) [26]. We found that mutation of the Gly202 andArg209 residues resulted in reduction in DNA-binding activity (Spo11m2) (Figure 4), suggesting that they are important for DNA-binding. However, mutation of the conserved Tyr88 did not affect DNA binding (Spo11m1) (Figure 4), suggesting that it is not involved in DNA binding. Furthermore, Spo11m1- and Spo11m2-expressed cell lines exhibited decrease of the levels of CWP1 protein, cyst generation, *cwp1-3,* and *myb2* mRNA (Figure 3 and Appendix A). Therefore, these specific regions for DNA binding or cleavage may be involved in activation of transcription, indicating a correlation of DNA binding or cleavage activity and in vivo function. RNA-seq analysis revealed a list of Spo11-affected genes (Appendix A). The decrease in tolerance to G418-mediated cytotoxicity in the Spo11td cell line (Figure 7D) suggests that Spo11 may affect many genes for survival in antibiotic stress.

Our findings provide new insights into the evolution of eukaryotic topoisomerase-related proteins from primitive to more complex eukaryotic cells, leading to greater understanding in the mechanism regulating gene transcription and cell differentiation for parasite transmission.

## 4. Materials and Methods

### 4.1. G. lamblia Culture

Trophozoites of *G. lamblia* WB, clone C6 (see ATCC 50803) (obtained from ATCC), were cultured in modified TYI-S33 medium [64]. Encystation was performed as previously described [14]. In experiments exposing *Giardia* vegetative trophozoites to G418, Spo11td and control cell line were cultured in growth medium at a beginning density of 1 × 10^6^ cells/mL with 518 μM G418.

### 4.2. Cyst Count

Cyst count was performed on the stationary phase cultures (~2 × 10^6^ cells/mL) during vegetative growth as previously described [65]. Cyst count was also performed on 24 h encysting cultures. Total cysts including both type I and II cysts [66] were counted in a hemacytometer chamber.

### 4.3. Isolation and Analysis of the spo11 Gene

Synthetic oligonucleotides used are shown in Appendix A. One putative homologue for Spo11, which is also annotated as Type II DNA topoisomerase VI subunit A (AF485825.1, orf 15279) was found in the *G. lamblia* genome database [8]. The Spo11 coding region with 300 bp of 5′- flanking region was cloned and sequenced. We also performed RT-PCR with *spo11*-specific primers using total RNA from *G. lamblia* to isolate its cDNA as previously described [20]. The cDNA was used as a template in subsequent PCR with primers spo11F and spo11R. Genomic and RT-PCR products were cloned into pGEM-T easy vector (Promega Corporation, Madison, WI, USA) and sequenced (Applied Biosystems, ABI, Foster City, CA, USA) and the results indicated no introns in the *spo11* gene.

### 4.4. Genomic DNA Extraction, PCR, and Quantitative Real-Time PCR Analysis

Synthetic oligonucleotides used are shown in Appendix A. Genomic DNA was isolated from trophozoites using standard procedures as previously described [54,67]. PCR analysis of *spo11* (AF485825.1, orf 15279), *cwp1* (U09330, orf 5638), *cwp2* (U28965, orf 5435), and *ran* (U02589, orf 15869) genes was performed using primers spo11F (PCR1F) and spo11R (PCR1R), PCR2F and PCR2R, cwp1F and cwp1R, cwp2F and cwp2R, ranF and ranR, respectively. Quantitative real-time PCR was conducted as previously described [54]. Specific primers were designed for detection of the *spo11*, *cwp1*, *cwp2*, and *ran* genes: spo11realF and spo11realR; cwp1realF and cwp1realR; cwp2realF and cwp2realR; and ranrealF and ranrealR.

### 4.5. RNA Extraction, RT-PCR, and Quantitative Real-Time PCR Analysis

Synthetic oligonucleotides used are shown in Appendix A. Total RNA was extracted and RT-PCR was performed as previously described [54]. The cDNA was used as a template in subsequent PCR. Semi-quantitative RT-PCR analysis of *spo11* (AF485825.1, orf 15279), *spo11-ha*, *cwp1* (U09330, orf 5638), *cwp2* (U28965, orf 5435), *cwp3* (AY061927, orf 2421), *myb2* (AY082882, orf 8722), *ran* (U02589, orf 15869), and 18S ribosomal RNA (M54878, orf r0019) gene expression was performed using primers spo11F and spo11R, spo11F and HAR, cwp1F and cwp1R, cwp2F and cwp2R, cwp3F and cwp3R, myb2F and myb2R, ranF and ranR, and 18SrealF and 18SrealR, respectively. Quantitative real-time PCR was performed as previously described with specific primers for detection of the *spo11*, *spo11-ha*, *cwp1*, *cwp2*, *cwp3*, *myb2*, *ran*, and 18S ribosomal RNA genes: spo11realF and spo11realR; spo11F and HAR; cwp1realF and cwp1realR; cwp2realF and cwp2realR; cwp3realF and cwp3realR; myb2realF and myb2realR; ranrealF and ranrealR; and 18SrealF and 18SrealR [54].

### 4.6. Plasmid Construction

Synthetic oligonucleotides used are shown in Appendix A. All constructs were verified by DNA sequencing as previously described [54]. Plasmid 5′Δ5N-Pac was a gift from Dr. Steven Singer and Dr. Theodore Nash [52]. Plasmid pgCas9 has been described previously [54]. To make construct pPSpo11, the *spo11* gene and its 300 bp of 5′- flanking region were amplified with oligonucleotides spo11NF and spo11MR, digested with NheI and MluI, and cloned into NheI and MluI digested pPop2NHA [68]. To make construct pPSpo11m1, or pPSpo11m2, the *spo11* gene was amplified using two primer pairs spo11m1F, or spo11m2F, and spo11MR, and spo11m1R, or spo11m2R and spo11NF. The two PCR products were purified and used as templates for a second PCR. The second PCR also included primers spo11NF and spo11MR, and the product was digested with Nhe and MluI and cloned into the NheI and MluI digested pPop2NHA [68].

The 700-bp 5′-flanking region of the *spo11* gene was amplified with oligonucleotides spo115HF and spo115NR, digested with *Hind*III/*Nco*I and cloned into *Hind*III/*Nco*I digested 5′Δ5N-Pac, resulting in spo115. The 700-bp 3′-flanking region of the *spo11* gene was amplified with oligonucleotides spo113XF and spo113KR, digested with *Xho*I/*Kpn*I and cloned into *Xho*I/*Kpn*I digested spo115, resulting in spo1153. We used gene synthesis services from IDT to obtain the fragment spo11-guide. The NCBI Nucleotide Blast search was used to avoid the potential off-target effects of guide sequence. The spo11-guide was digested with *Kpn*I/*Eco*RI and cloned into *Kpn*I/*Eco*RI digested spo1153, resulting in pSpo11td.

### 4.7. Transfection and Western Blot Analysis

Cells transfected with the pP series plasmids containing the *pac* gene were selected and maintained with 54 μg/mL (100 μM) of puromycin as previously described [52]. For CRISPR/Cas9 system, *Giardia* trophozoites were transfected with plasmid pSpo11td and pgCas9, and then selected in 100 μM puromycin as previously described [54]. The culture medium in the first replenishment contained 6 μM Scr7 and 100 μM puromycin as previously described [54]. The Spo11td stable transfectants were established after selection with puromycin. Stable transfectants were maintained at 100 μM puromycin and were further analyzed by Western blotting, or DNA/RNA extraction. The replacement of the *spo11* gene with the *pac* gene was confirmed by PCR and sequencing. The control is *G. lamblia* trophozoites transfected with double amounts of 5′Δ5N-Pac plasmid and selected with puromycin as previously described [54]. Puromycin was then removed from the medium for each stable cell line to obtain Spo11td –pu, and control –pu cell lines [54]. Subsequent analysis was performed after the removal of the drug for 1 month.

Western blots were probed with anti-V5-HRP (Invitrogen, Rockford, IL, USA), anti-HA monoclonal antibody (1/5000 in blocking buffer; MilliporeSigma, Burlington, MA, USA), anti-Spo11 (1/5000 in blocking buffer) (see below), anti-CWP1 (1/10000 in blocking buffer) [18], anti-MYB2 (1/5000 in blocking buffer) [21], anti-WRKY (1/5000 in blocking buffer) [19], anti-ISCS (1/10000 in blocking buffer) [25], anti-RAN (1/10000 in blocking buffer) [22], or preimmune serum (1/5000 in blocking buffer), and detected with HRP-conjugated goat anti-mouse IgG (1/5000; Pierce, Thermo Fisher Scientific Inc., Rockford, IL, USA) or HRP-conjugated goat anti-rabbit IgG (1/5000; Pierce, Thermo Fisher Scientific Inc., Rockford, IL, USA), and enhanced chemiluminescence (Merck Millipore, MilliporeSigma, Burlington, MA, USA). 

### 4.8. Expression and Purification of Recombinant Spo11 Protein

The genomic *spo11* gene was amplified using oligonucleotides Spo11F and Spo11R. The product was cloned into the expression vector pET101/D-TOPO (Invitrogen) in frame with the C-terminal His and V5 tags to generate plasmid pSpo11. To make the pSpo11m1, or pSpo11m2 expression vector, the *spo11* gene was amplified using primers spo11F and spo11R and specific template, including pPSpo11m1, or pPSpo11m2, and cloned into the expression vector. The pSpo11m1, or pPSpo11m2 plasmid was transformed into *Escherichia coli* and purified as previously described [25]. Protein purity and concentration were estimated by Coomassie Blue and silver staining compared with serum albumin. Spo11, Spo11m1, or Spo11m2 was purified to apparent homogeneity (>95%).

### 4.9. Generation of Anti-Spo11 Antibody

Rabbit anti-Spo11 polyclonal antibody was generated using purified Spo11 protein through a commercial vendor (GeneTex, Hsinchu City, Taiwan, ROC).

### 4.10. Immunofluorescence Assay

The pPSpo11, pPSpo11m1, or pPSpo11m2 stable transfectants cultured in growth medium or encystation medium for 24 h were harvested and subjected to immunofluorescence assay as previously described [9]. Anti-HA monoclonal antibody (1/300 in blocking buffer; Thermo Fisher Scientific, Waltham, MA, USA) and anti-mouse ALEXA 488 (1/500 in blocking buffer, Thermo Fisher Scientific, Waltham, MA, USA) were used as the detector.

### 4.11. Electrophoretic Mobility Shift Assay

Binding reactions with double-stranded plasmids or single-stranded DNA specified in the text were performed as previously described [15]. The pBluescript II SK (+) plasmid probe (150 fmol/rxn) was incubated with 0, 25, 50, 100, 200, and 400 nM of purified Spo11 at room temperature for 15 min. Glycerol was added to the reaction mixture to a 10% final concentration, and the mixture was separated in a 0.9% agarose gel by electrophoresis.

### 4.12. Co-Immunoprecipitation Assay

The stable cell lines cultured in growth medium were inoculated into encystation medium with puromycin and harvested and lysed after 24 h as previously described [23]. The cell lysates were incubated with anti-HA antibody conjugated to beads as previously described [23]. For reciprocal immunoprecipitation experiments, anti-WRKY or anti-MYB2 was used to do immunoprecipitation. The lysates were incubated with 2 μg of anti-WRKY, anti-MYB2 antibody or preimmune serum for 2 h and then incubated with protein G plus/protein A-agarose (Merck Millipore, MilliporeSigma, Burlington, MA, USA) for 1h. Proteins from the beads were analyzed by Western blotting using anti-HA monoclonal antibody (1/5000 in blocking buffer; MilliporeSigma, Burlington, MA, USA), or anti-WRKY (1/5000 in blocking buffer) [19], anti-MYB2 (1/5000 in blocking buffer) [21], anti-ISCS (1/10,000 in blocking buffer) [25], as previously described [23].

### 4.13. ChIP Assays

The pPSpo11 stable cell line and 5′Δ5N-Pac control cell lines cultured in growth medium were inoculated into encystation medium and harvested after 24 h and the assay was performed as previously described [23,25]. The chromatin extracts were incubated with anti-HA antibody conjugated to beads (MilliporeSigma, Burlington, MA, USA), as previously described [18]. The beads were washed and then incubated with elution buffer as previously described [25]. DNA was purified and subjected to PCR reaction followed by agarose gel electrophoresis or to quantitative real-time PCR as previously described [25]. Primers 18S5F and 18S5R were used to amplify the *18S ribosomal RNA* gene promoter as a control for our ChIP analysis. Primers spo115F and spo115R, cwp15F and cwp15R, cwp25F and cwp25R, cwp35F and cwp35R, myb25F and myb25R, and U65F and U65R were used to amplify *spo11*, *cwp1*, *cwp2*, *cwp3*, *myb2*, and *U6* gene promoters within the −200 to −1 region. 

### 4.14. RNA-Seq Analysis

Total RNA was isolated from the pPSpo11 and 5′△5N-Pac cell lines using TRIzol^®^ reagent (Thermo Fisher Scientific, Waltham, MA, USA). RNA concentration and quality were measured in a ND-1000 spectrophotometer (Nanodrop Technology, Thermo Fisher Scientific, Waltham, MA, USA) and analyzed in a Bioanalyzer 2100 (Agilent Technology, USA) with an RNA 6000 Nano labchip kit (Agilent Technologies, Santa Clara, CA, USA). Libraries for RNA sequencing were prepared using TruSeq RNA Sample Prep Kits v2 and then a single end sequencing (50SE) was performed on a Solexa platform (Illumina, San Diego, CA, USA). Sequencing was performed using sequencing-by-synthesis technology via the TruSeq SBS Kit (Illumina, San Diego, CA, USA). Illumina GA Pipeline software CASAVA v1.8 was used to obtain the reads and expected to generate 10 million reads per sample. Reads with low-quality scores were trimmed or removed using ConDeTri method [69], and then analyzed using TopHat/Cufflinks [70] for gene expression. FPKM (Fragments Per Kilobase of transcript per Million mapped reads) values were calculated to present expression levels. Statistical analysis was implemented using the CummeRbund package. Reference gene annotation were fetched from Ensembl database. The data has been deposited in the Gene Expression Omnibus (GEO) at NCBI with the accession number GSE185219.

## Figures and Tables

**Figure 1 ijms-22-11902-f001:**
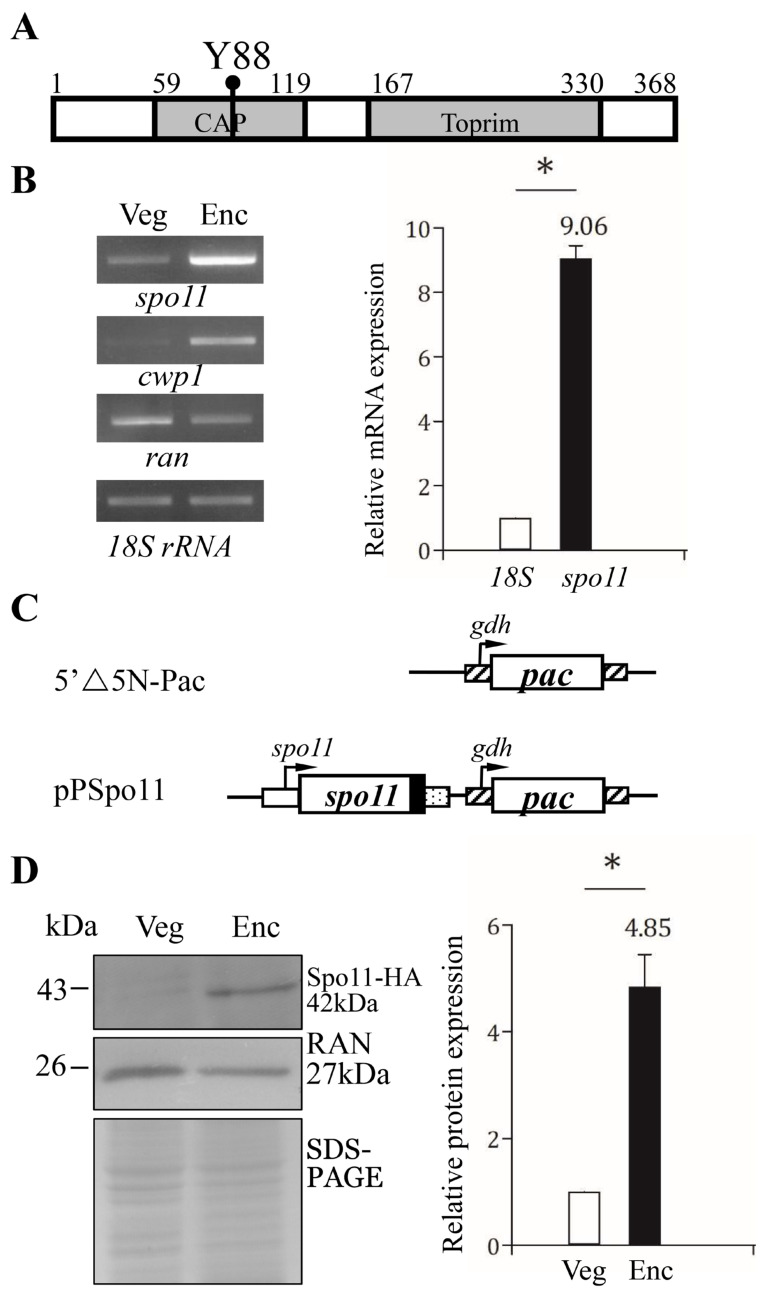
Spo11 protein and mRNA expression. (**A**) Schematic representation of the *Giardia* Spo11 protein. The gray boxes indicate the CAP and Toprim domains, as predicted by pfam. The conserved Tyr 88 (Y88) is indicated. (**B**) Spo11 mRNA level increased during encystation. RNA was extracted from *G. lamblia* wild type non-transfected WB cells incubated in growth medium (Veg, vegetative growth) or encystation medium for 24 h (Enc, encystation). RT-PCR was performed using primers for genes indicated in the figure (left panel). The mRNA levels of the *cwp1* and *ran* genes significantly increased and decreased during encystation, respectively. Real-time PCR was performed using primers for genes indicated in the figure (right panel). The mRNA levels were normalized to the 18S ribosomal RNA level. The ratio of mRNA levels in Enc sample to levels in Veg sample is shown and expressed as the means ± 95% confidence intervals of at least three separate experiments. *, *p* < 0.05. (**C**) Schematic presentation of the 5′∆5N-Pac and pPSpo11 plasmid. The *pac* gene (open box) is flanked by the 5′- and 3′-untranslated regions of the glutamate dehydrogenase (*gdh*) gene (striated box). In construct pPSpo11, the *spo11* gene is flanked by its own 5′-untranslated region (open box) and the 3′- untranslated region of the *ran* gene (dotted box). The coding sequence of the HA epitope tag is shown as a filled black box. (**D**) Spo11-HA protein level increased during encystation. The pPSpo11 stable transfectants were incubated in growth medium (Veg, vegetative growth) or encystation medium for 24 h (Enc, encystation) and then subjected to Western blot analysis using anti-HA and anti-RAN antibodies, respectively. SDS-PAGE with Coomassie Blue staining is included as a control for equal protein loading. The level of RAN protein was slightly decreased in encystation sample. The band intensity from triplicate Western blots was quantified using Image J. The Spo11 protein levels were normalized to the loading control (Coomassie Blue-stained proteins). The ratio of Spo11 protein levels in Enc sample to levels in Veg sample is shown and expressed as mean ± 95% confidence intervals. *, *p* < 0.05.

**Figure 2 ijms-22-11902-f002:**
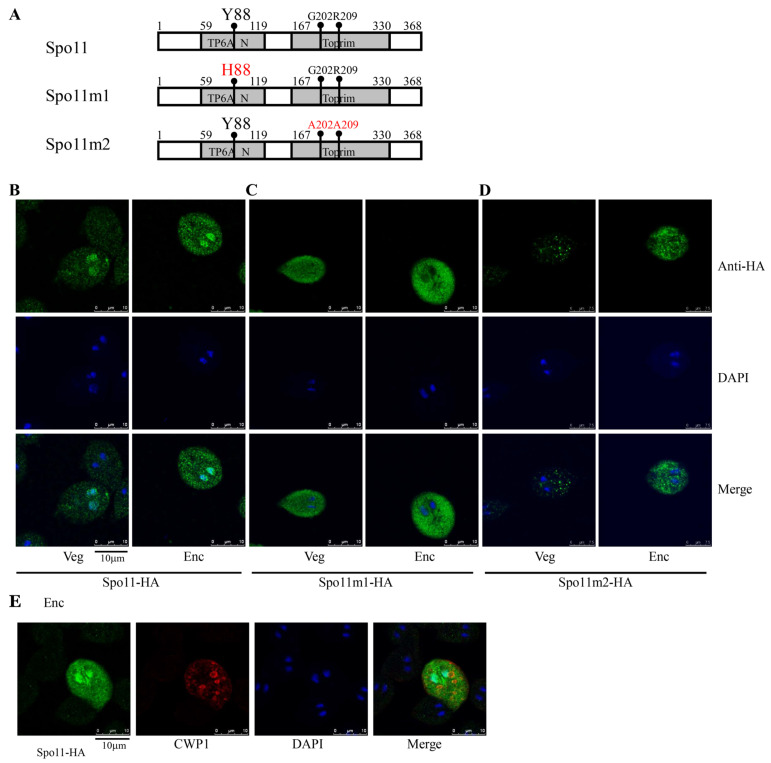
Localization of Spo11 and its mutants. (**A**) Diagrams of Spo11 and Spo11m1-2. The residue Tyr 88 (Y88), which is important for Spo11 cleavage activity, is mutated to His (H88) in Spo11m1. The residues Gly 202 (G202) and Arg 209 (R209), which are important for Spo11 DNA binding activity, are mutated to Ala (A202 and A209) in Spo11m2. The *spo11* gene was mutated and subcloned to replace the wild-type *spo11* gene in the backbone of pPSpo11 (Figure 1C), and the resulting plasmids pPSpo11m1-2 were transfected into *Giardia*. The expression cassettes of the *pac* gene and *spo11* gene are the same as in Figure 1C. (**B**) Localization of Spo11 protein. The pPSpo11 stable transfectants were incubated in growth medium (Veg, left panel) or encystation medium for 24 h (Enc, right panel), and immunofluorescence assays were performed using anti-HA antibody. The upper panels show the localization of the Spo11 protein. The middle and bottom panels show the DAPI staining of cell nuclei and the merged images, respectively. (**C**,**D**) Immunofluorescence analysis of Spo11m1-2 distribution. The pPSpo11m1-2 stable transfectants were cultured and then subjected to immunofluorescence analysis. (**C**) The products of pPSpo11m1 localized to the cytoplasm in both vegetative and encysting trophozoites. (**D**) The products of pPSpo11m2 localized to the vesicles in cytoplasm in vegetative trophozoites and to the cytoplasm with some vesicles in encysting trophozoites. (**E**) Localization of CWP1 in the Spo11-expressing cell line. The pPSpo11 stable transfectants were cultured in encystation medium for 24 h and then subjected to immunofluorescence assays. The endogenous CWP1 protein and vector-expressed Spo11-HA protein were detected by anti-CWP1 and anti-HA antibodies, respectively. The left panel shows that the Spo11-HA protein is localized to the nuclei and slightly to the cytoplasm. The middle panel shows that the CWP1 protein is localized to the ESVs. The right panel shows the merged image.

**Figure 3 ijms-22-11902-f003:**
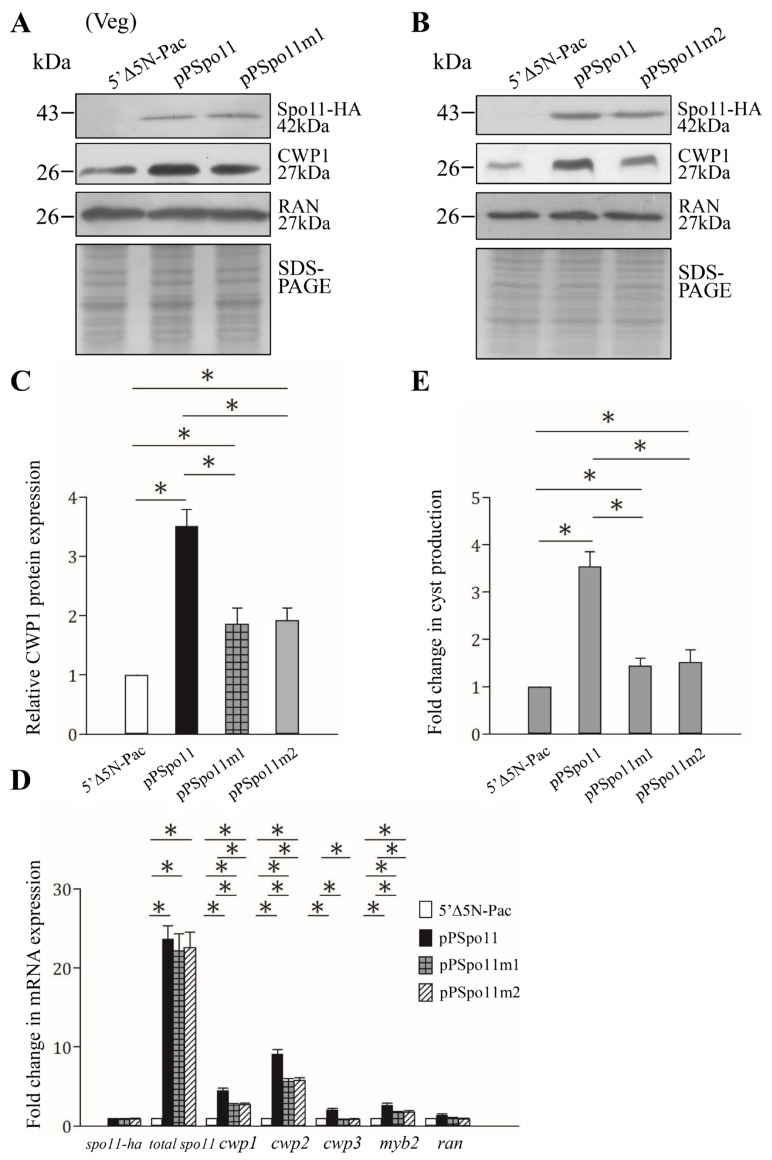
Increased expression of the *cwp1-3* and *myb2* genes in the Spo11-expressing cell line. (**A**,**B**) Spo11 expression increased the CWP1 level. The 5′∆5N-Pac, pPSpo11, pPSpo11m1 (**A**), and pPSpo11m2 (**B**) stable transfectants were incubated in growth medium and then subjected to Western blot analysis using anti-HA, anti-CWP1, and anti-RAN antibodies, respectively. SDS-PAGE with Coomassie Blue staining is included as a control for equal protein loading. As a control, similar levels of the RAN protein were detected. (**C**) Quantification of the band intensity from triplicate Western blots using Image J. The CWP1 protein levels were normalized to the RAN loading control. The ratio of CWP1 protein levels in the specific cell line to levels in the 5′∆5N-Pac cell line is shown and expressed as mean ± 95% confidence intervals. *, *p* < 0.05. (**D**) Quantitative real-time PCR assays of transcript expression in the Spo11- and Spo11m2-expressing cell lines. Real-time PCR was performed using primers for genes indicated in the figure. Similar levels of the 18S ribosomal RNA were detected. The mRNA levels were normalized to the 18S ribosomal RNA levels. The ratio of mRNA levels in the pPSpo11 or pPSpo11m2 cell line to levels in the 5′∆5NPac cell line is shown and expressed as the mean ± 95% confidence intervals of at least three separate experiments. *, *p* < 0.05. (**E**) Expression of Spo11 induced cyst generation. Cyst number was counted from the 5′∆5N-Pac, pPSpo11, and pPSpo11m2 stable transfectants cultured in growth medium as described in “Materials and Methods”. Fold changes in cyst generation are shown as the ratio of the sum of total cysts in the pPSpo11 or pPSpo11m2 cell lines relative to the 5′∆5NPac cell line. Values are shown as mean ± 95% confidence intervals. *, *p* < 0.05.

**Figure 4 ijms-22-11902-f004:**
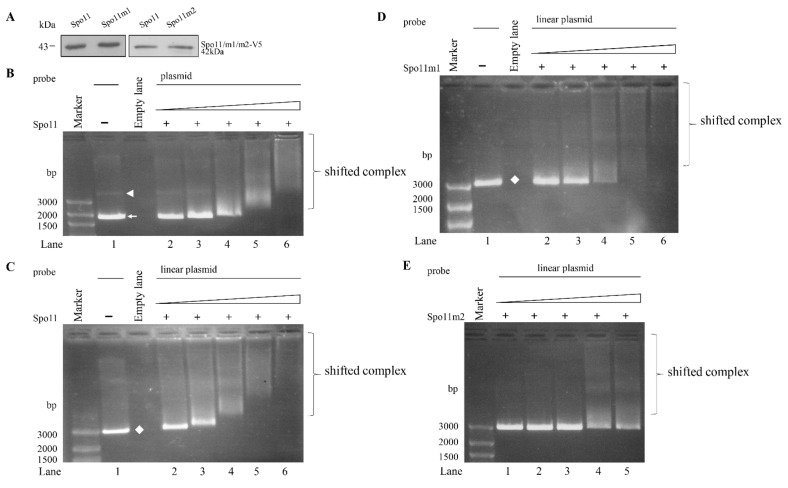
DNA-binding ability of Spo11 revealed by electrophoretic mobility shift assays with plasmid DNA probe. (**A**) Spo11 and its mutants were purified from *E. coli* and detected by Western blot using anti-V5 antibody. (**B**) Detection of DNA binding ability of Spo11 with circular plasmid DNA probe. Purified Spo11 and the pBluescript II SK(+) plasmid probe were used in electrophoretic mobility shift assays. Binding reaction mixtures contained the components indicated above the lanes. The plasmid probe (300 ng/rxn) was incubated with 26, 53, 105, 210, and 420 ng purified Spo11 (lanes 2 to 6, respectively). The white arrowhead and arrow indicate supercoiled circular plasmid DNA and relaxed circular plasmid DNA, respectively. (**C**) Detection of DNA binding ability of Spo11 with linear plasmid DNA probe. Purified Spo11 and the linear pBluescript II SK(+) plasmid DNA probe were used in electrophoretic mobility shift assays. Binding reaction mixtures contained the components indicated above the lanes. The probe (300 ng/rxn) was incubated with 26, 53, 105, 210, and 420 ng purified Spo11 (lanes 2 to 6, respectively). The white diamond indicates linear plasmid DNA. (**D**) DNA binding ability of Spo11m1 with linear plasmid DNA probe. Purified Spo11m1 and the linear pBluescript II SK(+) plasmid DNA probe were used in electrophoretic mobility shift assays. Binding reaction mixtures contained the components indicated above the lanes. The probe (300 ng/rxn) was incubated with 26, 53, 105, 210, and 420 ng purified Spo11m1 (lanes 2 to 6, respectively). The white diamond indicates linear plasmid DNA. (**E**) Decrease of DNA binding ability of Spo11m2 with linear plasmid DNA probe. Purified Spo11m2 and the linear pBluescript II SK(+) plasmid DNA probe were used in electrophoretic mobility shift assays. Binding reaction mixtures contained the components indicated above the lanes. The probe (300 ng/rxn) was incubated with 26, 53, 105, 210, and 420 ng purified Spo11m2 (lanes 1 to 5, respectively).

**Figure 5 ijms-22-11902-f005:**
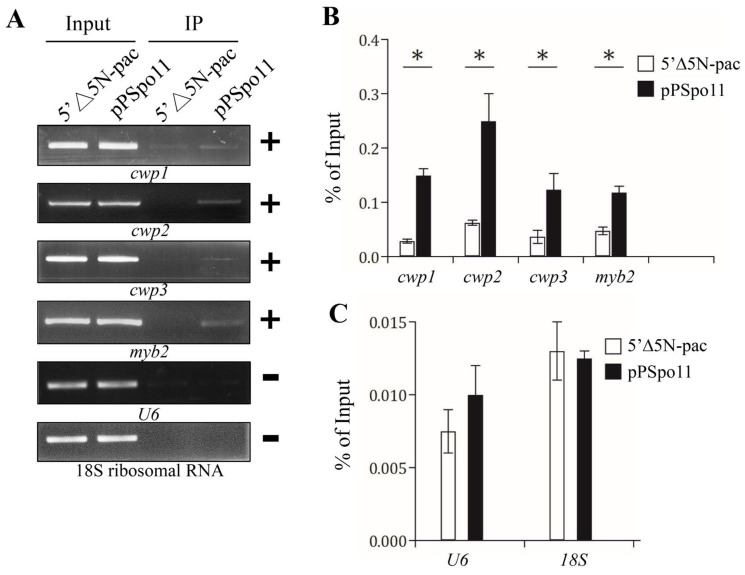
Recruitment of Spo11 to the *cwp1-3* and *myb2* promoters. (**A**) ChIP assays. The pPSpo11 cell line expressing the HA-tagged Spo11 and the 5‘∆5N-Pac cell line as the control cell line were cultured in encystation medium for 24 h and then subjected to ChIP assays. Anti-HA antibody conjugated to beads was used to assess interaction of Spo11-HA with various gene promoters. Immunoprecipitated chromatin was analyzed by PCR using primers that amplify the 5′-flanking region of the genes indicated in the figure. Approximately the same amount of PCR product was obtained from input DNA in the pPSpo11 cell line and the control cell line. At least three independent experiments were performed. Representative results are shown. Immunoprecipitated products of Spo11-HA yielded more PCR products of *cwp1*, *cwp2*, *cwp3*, and *myb2* promoters in the pPSpo11 cell line relative to the control cell line, indicating that Spo11-HA was bound to these promoters (+). Spo11-HA was not bound to the U6 promoter fragment (–). The *18S ribosomal RNA* gene promoter was used as a negative control (–). (**B**,**C**) ChIP assays coupled with quantitative PCR. Values represented as a percentage of the antibody-enriched chromatin relative to the total input chromatin (% of Input). Results are expressed as the mean ± 95% confidence intervals of at least three experiments. *, *p* < 0.05.

**Figure 6 ijms-22-11902-f006:**
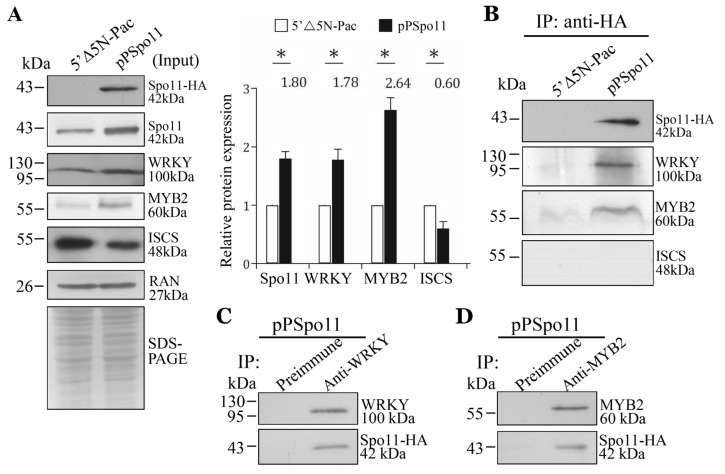
Interaction between Spo11 and WRKY, or MYB2. (**A**) Expression of the HA-tagged Spo11 protein in whole cell extracts for co-immuno-precipitation assays (Input). The 5′∆5N-Pac and pPSpo11 stable transfectants were cultured in encystation medium for 24 h. SDS-PAGE and Western blot were performed as described in Figure 3A. The blot was probed with anti-HA, anti-Spo11, anti-WRKY, anti-MYB2, anti-ISCS, and anti-RAN antibodies, respectively. The band intensity from triplicate Western blots was quantified using Image J as described in Figure 3A. (**B**) Interaction between Spo11 and WRKY, or MYB2 was detected by co-immuno-precipitation assays. The 5′∆5N-Pac and pPSpo11 stable transfectants were cultured in encystation medium for 24 h. Proteins from cell lysates were immuno-precipitated using anti-HA antibody and analyzed by Western blotting with anti-HA, anti-WRKY, anti-MYB2, and anti-ISCS antibodies, respectively. (**C**,**D**) Reciprocal immuno-precipitation for confirmation of Spo11 and (C) WRKY, or (D) MYB2 interaction. The pPSpo11 stable transfectants were cultured in encystation medium for 24 h. Proteins from cell lysates were immuno-precipitated using (**C**) anti-WRKY or (**D**) anti-MYB2 antibody, or preimmune serum (negative control) and analyzed by Western blotting with (**C**) anti-WRKY, or (**D**) anti-MYB2 and anti-HA antibodies, respectively.

**Figure 7 ijms-22-11902-f007:**
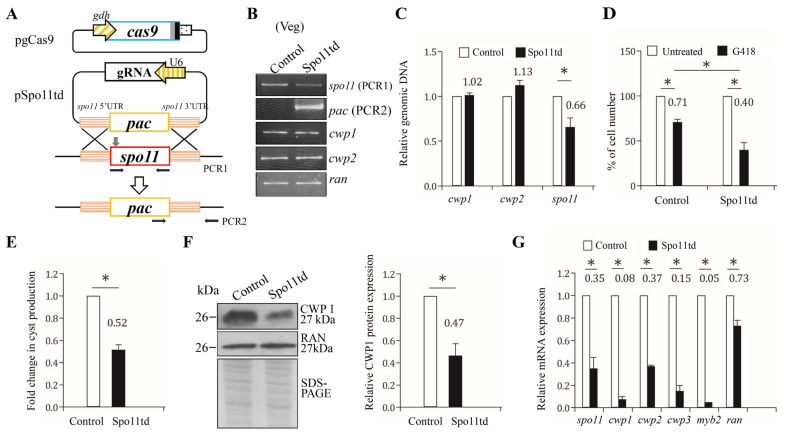
Targeted disruption of the *spo11* gene resulted in decreased expression of the *cwp1-3* and *myb2* genes during vegetative growth. (**A**) Schematic presentation of the pgCas9 and pSpo11td plasmids. In construct pgCas9, the *cas9* gene is flanked by *gdh* promoter (striated box) and 3′ untranslated region of the *ran* gene (dotted box). The nuclear localization signal (filled gray box) and an HA tag (filled black box) are fuse to the C terminus. In construct pSpo11td, a single gRNA is under the control of the *Giardia* U6 promoter. The single gRNA, which is located upstream three nucleotides of protospacer-adjacent motif (NGG sequence), includes a guide sequence targeting 20-nucleotide of the *spo11* gene (nt 262–281). pSpo11td also has the HR template cassette composed of the *pac* selectable marker and the 5′ and 3′ flanking region of the *spo11* gene as homologous arms. The Cas9/gRNA cutting site in the genomic *spo11* gene is indicated by a gray arrow. Replacement of the genomic *spo11* gene with the *pac* gene will occur by HR after introducing a DSB in the *spo11* gene. After transfection of the pgCas9 and pSpo11td constructs into *G. lamblia* WB trophozoites, the Spo11td stable transfectants were established under puromycin selection. The control cell line is trophozoites transfected with double amounts of 5′Δ5N-Pac plasmid (Figure 2C) and selected with puromycin. PCR1/2 were used for identification of clones with targeted disruption. (**B**) PCR confirmed partial replacement of the *spo11* gene with the *pac* gene in the Spo11td cell line. Genomic DNA was isolated from the Spo11td and control cell lines cultured in growth medium with puromycin (vegetative growth, Veg) and subjected to PCR using primers specific for *spo11* (PCR1 in panel A), *pac* (PCR2 in panel A), *cwp1*, *cwp2*, and *ran* genes, respectively. Products from the *cwp1*, *cwp2*, and *ran* genes are internal controls. (**C**) Real-time PCR confirmed partial disruption of the *spo11* gene in the Spo11td cell line. Real-time PCR was performed using primers specific for *spo11*, *cwp1*, *cwp2*, and *ran* genes, respectively. The *spo11*, *cwp1*, and *cwp2* DNA levels were normalized to the *ran* DNA level. The ratio of DNA levels in Spo11td cell line to levels in control cell line is shown and expressed as the means ± 95% confidence intervals of at least three separate experiments. *, *p* < 0.05. (**D**) G418 sensitivity increased by targeted disruption of the *spo11* gene. The Spo11td and control cell lines were subcultured in growth medium containing 518 μM G418 for 24 h and then subjected to cell count. An equal volume of ddH2O was added to cultures as a negative control. Fold changes in cell number are shown as the ratio of cell number in the G418 sample relative to the ddH2O sample. Values are shown as mean ± 95% confidence intervals of three independent experiments. *, *p* < 0.05. (**E**) Targeted disruption of the *spo11* gene in the Spo11td cell line resulted in decreased cyst generation during vegetative growth. Cyst number was counted from the control and Spo11td cell lines cultured in growth medium as described in “Materials and Methods” and Figure 3D. (**F**) The CWP1 level decreased by targeted disruption of the *spo11* gene in the Spo11td cell line during vegetative growth. The control and Spo11td cell lines cultured in growth medium were subjected to SDS-PAGE and Western blot analysis as described in Figure 3A. The blot was probed with anti-CWP1 and anti-RAN antibodies, respectively. The band intensity from triplicate Western blots was quantified using Image J as described in Figure 3A. (**G**) Targeted disruption of the *spo11* gene in the Spo11td cell line resulted in decreased expression of *cwp1-3* and *myb2* during vegetative growth. The control and Spo11td cell lines cultured in growth medium were subjected to quantitative real-time RT-PCR analysis using primers specific for *spo11*, *cwp1*, *cwp2*, *cwp3*, *myb2*, *ran*, and 18S ribosomal RNA genes, respectively, as described in Figure 2A.

**Figure 8 ijms-22-11902-f008:**
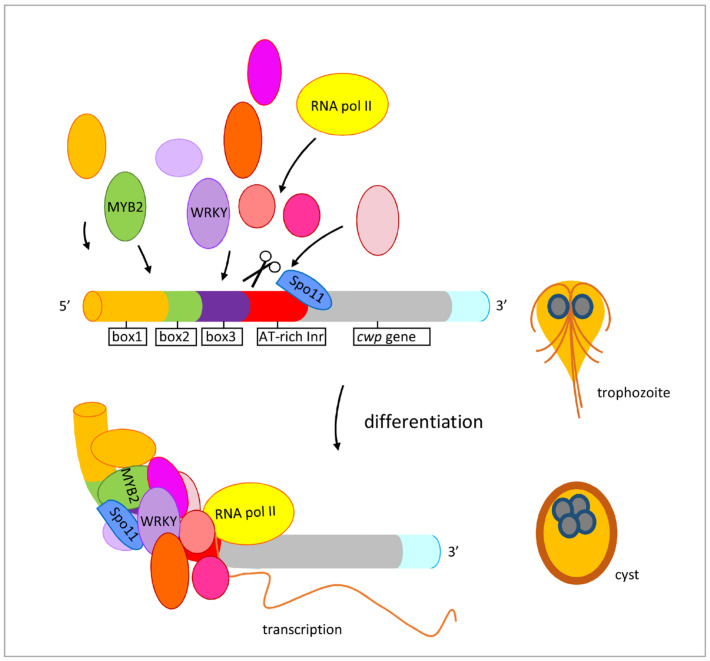
Induced expression of encystation-induced *cwp1-3* genes by Spo11 in cyst differentiation process. The *cwp1-3* genes encoding key construction component of the cyst wall are up-regulated by Spo11, which recruits MYB2 and WRKY transcription factors in cyst differentiation process. These transcription factors can bind to *cis*-acting elements of the *cwp* promoters, such as box1-3 or AT-rich initiator (Inr), to activate *cwp* gene transcription. They can form complexes to recruit RNA polymerase II for transcription initiation. CWP1 is present in vegetative trophozoite stage at a lower level and displays higher expression levels by these transcription factors in cyst differentiation process. The increase of Spo11, WRKY, MYB2, and other transcription factors during encystation may further induce CWP1 expression, resulting in more cyst generation.

## Data Availability

Publicly available datasets were analyzed in this study. This data can be found here: https://www.ncbi.nlm.nih.gov/geo/query/acc.cgi?acc=GSE185219 (accessed on 21 October 2021).

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
