# Peer review of "A Novel Spo11 Homologue Functions as a Positive Regulator in Cyst Differentiation in Giardia lamblia"

_ijms, 2021, doi:10.3390/ijms222111902_

Round 1

Reviewer 1 Report

Comments

 A Novel Spo11 Homologue Functions as A Positive Regulator 2 in Cyst Differentiation in Giardia Lamblia

In this manuscript the authors proposed a role for a Spo11 homologue, which is known to be responsible by formation of double-stranded breaks by a topoisomerase-like Spo11. They proposed that Spo11 can activate encystation process, like other topoisomerases they have previously characterized. They found that Spo11 was capable of binding to both single-stranded and double-stranded DNA in vitro and it could also bind to the cwp promoters in vivo as accessed in chromatin immunoprecipitation assays. After stable transfection with the plasmid, they found the expression of the Spo11-HA protein significantly increased during encystation They suggest that Spo11 acts as a positive regulator for Giardia differentiation into cyst.

 Despite being an important topic of interest, some review is needed.

This referee makes the following comments on some points of this manuscript, which needs improvement.

Title: Please, do not use capital letter in the species name. The correct is Giardia lamblia

The authors stated that there is “ exchange of genetic material in the fused nuclei occurs during Giardia encystation, suggesting sexual processes of this protozoan”- This has not yet been fully proven.

Introduction:

Lines 4-5: “of synthesis of three cyst wall proteins (CWPs) transported with polysaccharide is a hallmark of cyst wall formation”.

Please change this part of the text. The carbohydrates that form the cyst wall come from a vesicle separate from the vesicles that carry the protein part. Authors should be updated in the literature as they do not correctly cite the origin of the wall, the encystment vesicles (ESVs) and carbohydrates.

In addition, this referee does not understand why the cited references are placed in numbers and also by authors' names in the text. Why this duplicity? Journal model requirement?

Fig. 4 Legend: Please correct the word plasmid

Reviewer 2 Report

Dear Editors and authors

This manuscript studied the targeted disruption of the spo11 gene with the CRISPR/Cas9 system, the results suggest that Spo11 acts as a positive regulator for Giardia differentiation into a cyst. This is a good study of the Spo11 function other than meiotic recombination in protozoan parasites. Importantly, the authors' experimental data confirmed their hypothesis. Generally, the manuscript is well written, but there are still some English grammar problems. It is best to find a native English speaker to improve the level of English.

Comments and suggestions:

There should be spaces between numbers and units throughout the manuscript.

Abstract

Lines 13 and 20: the first appearance should also show the full names of Small, PERKY, and MYB2, not just abbreviation of the name.

Line 25: change leaded to led.

Introduction

Lines 8, 9, and 13: I think it is better to also show the full names of Myb2, GARP1, 8 ARID1, WRKY, E2F1, Pax1, Pax2, MBF1, TOP2 or TOP3, and Spoll.

Line 33: change function to functions.

Result

Please remake Figure 1B and 1D, Figure 3D, and Figure 5A, not neat, to be difficult to understand immediately.

Please delete the small number on pages 4 and 5.

Discussion

It is not good to put all the Figure names into the discussion part. Such as, on pages 13—lines 49-50, "In vivo association of Spo11 with the cwp1-3 and myb2 promoters was also confirmed (Figure 6). In addition, Spo11 can be co-immunoprecipitated with WRKY and Myb2 (Figure 5).", please delete “(Figure 6)” and “(Figure 5)”.
